# Influence of inter-fractional respiratory motion changes on dose delivery accuracy in dynamic conformal arc lung stereotactic body radiotherapy: A phantom study

**Bong Kyung Bae**[1], **Sung Joon Kim**[2], **Jae-Chul Kim**[1]*

1 Department of Radiation Oncology, School of Medicine, Kyungpook National University, Daegu, Korea,
2 Department of Radiation Oncology, Kyungpook National University Chilgok Hospital, Daegu, Korea

* jckim@knu.ac.kr

## Abstract

### Purpose

To evaluate the influence of inter-fractional respiratory motion variation on dose delivery accuracy in dynamic conformal arc lung stereotactic body radiotherapy (SBRT) using glass dosimeter and QUASAR™ respiratory motion phantom.

### Materials and Methods

Four-dimensional computed tomography (4D-CT) was acquired using a sinusoidal respiratory waveform (amplitude: 10 mm, breaths per minute [BPM]: 20). Three glass dosimeters were positioned at the superior edge, geometric center, and inferior edge of the tumor target. Internal target volume (ITV)-SBRT and gated-SBRT plans were created and delivered under nine respiratory conditions (BPM: 10–30, amplitude: 5–30 mm). Treatment was considered acceptable if the delivered dose to the glass dosimeters remained within the $D_{5\%}$–$D_{95\%}$ of the gross tumor volume.

### Results

BPM had minimal effect on ITV-SBRT, with the doses delivered to the target remaining within the acceptable range for all BPMs. However, amplitude significantly affected SBRT accuracy. For ITV-SBRT, increase in amplitude caused underdose at both the superior and inferior edges. In gated-SBRT, higher amplitude led to significant underdosing at superior edge of the target than that observed in ITV-SBRT, while inferior edge remained within the acceptable dose range. Underdose worsened with increasing amplitude, and 10 mm increase from the reference caused it to fall below the acceptable range ($D_{95\%}$).

**Data availability statement:** All relevant data are within the paper and its Supporting Information files.

**Funding:** The author(s) received no specific funding for this work.

**Competing interests:** The authors have declared that no competing interests exist.

## Conclusion

Respiratory motion significantly affects dose delivery accuracy in lung SBRT, with amplitude playing a critical role. An amplitude increase of ≥ 10 mm from CT acquisition during SBRT delivery resulted in a significant target underdosing below the clinically acceptable threshold.

## Introduction

Stereotactic body radiotherapy (SBRT) is an advanced radiation therapy (RT) technique, delivers highly conformal doses to small target lesions while minimizing exposure to surrounding normal tissues [1]. In lung SBRT, a biologically effective dose (BED) of ≥ 100 Gy is generally administered, resulting in favorable local control [2–5]. Consequently, SBRT has emerged as an viable treatment option for early-stage non-small cell lung cancer and is widely adopted in clinical practice [6,7].

Considering that SBRT delivers large ablative doses to small targets, precise targeting is critical. To achieve precise delivery, various image guidance and motion-management techniques are employed [8,9]. Lung tumors move with respiration, and due to the inherent variability in patient breathing, intra-fractional and inter-fractional variations in respiratory motion and tumor motion are inevitable [10]. These variations can significantly affect the precision of high-accuracy treatments such as SBRT. If not properly managed, such movements can lead to geographical miss and inaccurate dose distribution, potentially resulting in treatment failure or unintended toxicity [8,9,11].

Despite numerous studies have examined the effects on respiratory motion in lung SBRT, most studies have focused on treatment planning system (TPS) simulations or film dosimetry, comparing dose distributions between TPS calculations or film dosimetry [12–17]. While film dosimetry offers high spatial resolution for evaluating global dose distributions, accurately assessing the dose delivered to a specific point within a moving target is challenging due to motion-induced blurring and potential geometric uncertainties [18]. Point dosimetry, using detectors like glass dosimeters, can provide accurate dose measurement to a specific locations within the moving target [19]. However, studies evaluating the actual delivered dose using point dosimetry under varying inter-fractional motion conditions remain limited. Therefore, this study aims to quantitatively evaluate the influence of inter-fractional respiratory motion changes on lung SBRT dose delivery accuracy by measuring the dose at discrete points within a moving phantom target using glass dosimeters. Additionally, we seek to establish an acceptable range for respiratory variations based on dosimetric measurements.

### Materials and methods

#### Glass dosimeter and respiratory motion phantom

Radio-photoluminescent glass dosimeters (GD-302M; AGC Techno Glass, Shizuoka, Japan) were used to measure the radiation dose delivered to the tumor target within the phantom (Fig 1a). Each dosimeter element was engraved with

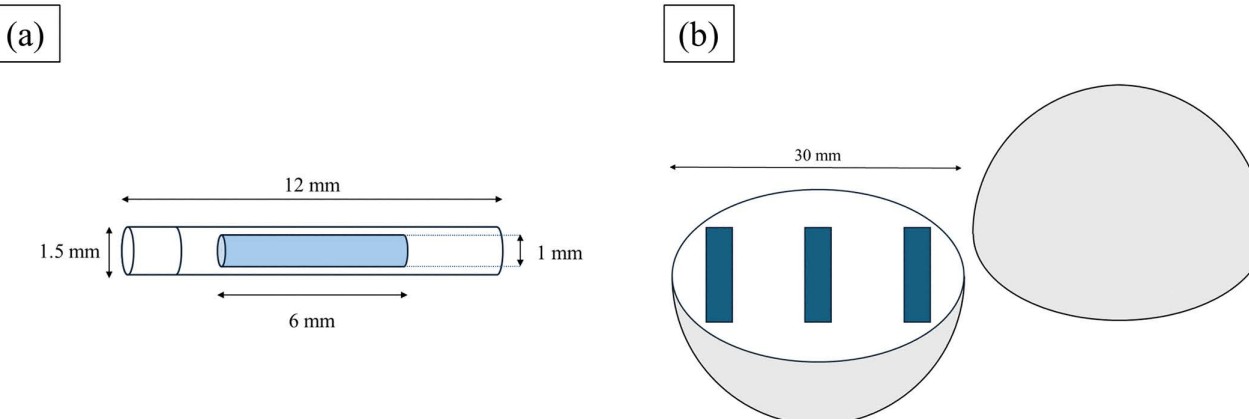

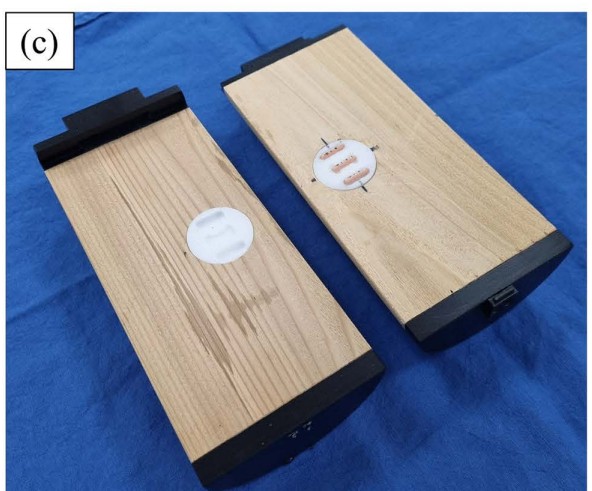
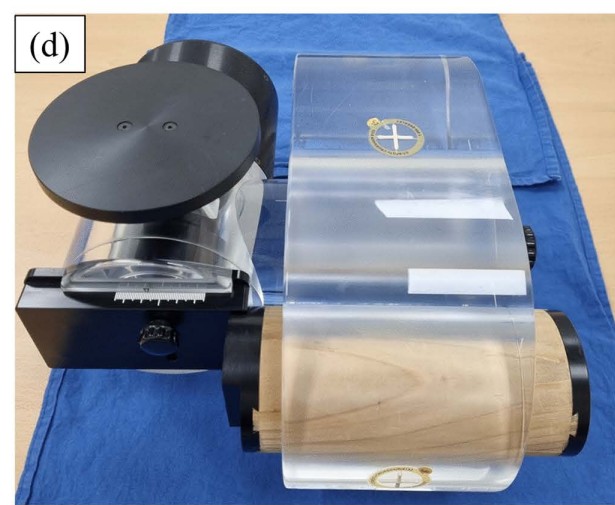

**Fig 1. Illustration of (a) radio-photoluminescent glass dosimeters (GD-302M, AGC Techno Glass) and (b) a custom-built, 30 mm diameter split-spherical Teflon target with carved grooves for glass dosimeter insertion. (c)** The target placed within an offset split-cedar lung insert. **(d)** The insert positioned within the QUASAR™ Respiratory Motion Phantom for CT acquisition and treatment delivery.

a unique identification (ID) number for tracking. The dosimeters measured 12 mm in length and 1.5 mm in diameter, with a readout volume of 6 mm in length and 1 mm in diameter. Their composition included 31.55% phosphorus (P), 51.16% oxygen (O), 6.12% aluminum (Al), 11.0% sodium (Na), and 0.17% silver (Ag). To read the dosimeters, a reading magazine (FGD-M151, AGC Techno Glass) and a compatible reader (FGD-1000SE, AGC Techno Glass) were used [20,21].

All glass dosimeters were annealed prior to irradiation using a small electric oven (Hi-Cera Kiln NHK-210-BS2, Nitto Kagaku, Japan) at 400 °C for 60 min, followed by cooling to below 40 °C for initialization. Standard calibration of the reader was performed using a manufacturer-supplied reference glass element. Calibration factors were determined individually for each glass element under the same static SBRT beam conditions as the experiment, thereby minimizing potential uncertainties such as angular dependence. The reproducibility of calibration factors was verified by repeated irradiations across a wide dose range, confirming the excellent linearity of glass dosimeter response. The energy

dependence of glass dosimeters is known to be minimal in the MV photon range, and since all irradiations in this study were performed with a single 6 MV FFF beam, energy dependence was not considered a significant factor.

Before irradiation, the initial background signal of each element (typically 10–30 µGy) was measured and subtracted from subsequent readouts. After irradiation, all elements were pre-heated at 70 °C for 30 min using a forced convection oven (DKN302, Yamato, Japan) to stabilize the radio-photoluminescence signal and minimize fading. Inter-dosimeter variability was managed by applying element-specific calibration factors rather than pooled calibration. All readout data were stored electronically, and the dosimeters were kept in a desiccator to prevent variability due to humidity [22].

A custom-manufactured Teflon tumor target with a 30 mm diameter was used as the tumor target within the phantom. Three dosimeters were placed inside the tumor target to assess dose delivery vulnerabilities to respiratory motion. One dosimeter was placed in the geometric center to verify the planned target dose. Two dosimeters were placed on just inside the superior and inferior edges of the target. While these locations are intended to receive a stable, high prescription dose under ideal conditions, their proximity to the steep dose fall-off region makes them the most sensitive indicators of under-dosing resulting from geometric misses due to respiratory motion(Fig 1b). The tumor target was inserted into an offset split cedar lung insert (Model 500–3333) supplied by the manufacturer of the QUASAR™ Respiratory Motion Phantom (Modus Medical Devices, London, Ontario, Canada) (Fig 1c).

The QUASAR™ Respiratory Motion Phantom, a programmable motorized system designed to replicate breathing motion based on user-defined inputs, was used in this study (Fig 1d). The phantom moved the insert in the superior-inferior direction based on the specified motion parameters. Respiratory waveforms were generated using the wave editor module of the QUASAR™ Programmable Respiratory Motion Software (Modus Medical Devices). A sinusoidal waveform with a 10 mm amplitude and a breathing rate of 20 breaths per minute (BPM) was created as the reference respiratory motion for four-dimensional computed tomography (4D-CT) acquisition. Seven additional sinusoidal waveforms were generated for radiotherapy delivery, incorporating two different breathing rates (10 BPM and 30 BPM) and five amplitudes (5 mm, 15 mm, 20 mm, 25 mm, and 30 mm) based on the reference respiratory motion. In total, nine respiratory waveforms were generated, including a static waveform.

While film dosimetry provides high spatial resolution, glass dosimeters were chosen for this study for precise dose measurements at specific pre-defined points (superior edge, geometric center, and inferior edge) within the moving target volume, mitigating motion-induced blurring often observed with film dosimetry [21]. Each glass dosimeter effectively integrates the dose delivered to its small volume as it moves with the Teflon tumor target.

## Computed tomography image acquisition

Two sets of computed tomography (CT) images were obtained using a Philips Brilliance CT Big Bore scanner (Philips Medical Systems, Cleveland, OH, USA) with a 3 mm slice thickness. A 4D-CT set was acquired for the reference respiratory motion (amplitude: 10 mm, BPM: 20). The Real-Time Position Management (RPM™) Respiratory Gating System (Varian Medical Systems, Palo Alto, CA, USA) was used to record respiratory motion data. The acquired CT images were sorted and binned into 10 phases using the phase-binning method. A static CT set was obtained for glass dosimeter factor correction. CT image sets were acquired with the tumor target and glass dosimeter in place to replicate actual treatment conditions.

## Tumor target delineation and radiotherapy planning

Gross tumor volume (GTV) was delineated for each respiratory phase and transferred to the corresponding average intensity projection (AIP) image sets. For full-phase internal target volume (ITV)-SBRT planning, GTVs from all respiratory phases were mapped onto the AIP image set encompassing all phases. For gated-SBRT planning, GTVs from the 30–70% respiratory phases were mapped onto the corresponding AIP image sets of those phases. The ITV was defined

as the sum of the respective GTVs. A 5-mm isotropic margin was applied to the ITV to generate the planning target volume (PTV). For static CT based SBRT planning, a 5-mm isotropic margin was directly applied to generate the PTV.

SBRT was planned and delivered using a Vital Beam system (Varian Medical Systems) with the dynamic conformal arc (DCA) technique. Dose calculation was performed using the AcurosXB (AXB, version 18.1) algorithm with resolution size of 2.5 mm. While finer grid resolutions are often recommended for SBRT to accurately model steep dose gradients, this grid size was deemed appropriate for the specific objectives of this study. The goal of this study is to evaluate relative dosimetric changes due to motion, for which this resolution provides sufficient accuracy. Furthermore, previous studies have reported that the dosimetric differences between 2.5 mm grid to finer grid are generally within 1–2%, which is less significant than the uncertainties introduced by respiratory motion and glass dosimeter measurements in current study [23,24]. No density overrides were applied within the PTV. The plan consisted of two partial arcs of 180 degrees using a 6 MV flattening filter free photon beams. The prescription dose was 5 Gy per fraction, following the RTOG 0813 and RTOG 0915 lung SBRT protocols [3,4]. The maximum dose was delivered to a point within the PTV and normalized to 100%. The prescription isodose surface ranged from ≥ 60% to < 90% of the maximum dose. Adequate target coverage required that 95% of the PTV receive the prescribed dose and that 99% of the PTV receive > 90% of the prescribed dose. A dose exceeding 105% of the prescription dose was restricted to the PTV, and the ratio of the prescription dose volume to the PTV was maintained < 1.2.

### Radiotherapy delivery and dosimetric analysis

SBRT was planned and delivered using static CT imaging to establish reference values for the respective glass dosimeters. The calibration factor for each dosimeter was calculated based on the absorbed dose, exposure dose, and background values obtained from the static SBRT plan and its corresponding delivery. The dose delivered to the glass dosimeters during SBRT with respiratory motion was calculated using the calibration factor for each dosimeter. S1 Table presents a summary of the calibration results.

ITV-SBRT and gated-SBRT plans were delivered across nine different respiratory motion patterns. Each SBRT delivery was repeated three times per respiratory motion to assess dose-delivery variations across treatment fractions. The delivered dose was evaluated using three glass dosimeters positioned within the tumor target at the superior edge, center, and inferior edge to analyze dose homogeneity. Glass dosimeter measurements were compared to the planned dose calculated by the TPS. Results are presented in absolute dose (cGy), supplemented by percentages relative to the mean GTV dose for respective SBRT plans when needed. Treatment delivery was considered acceptable if the dose measured by glass dosimeters was within the range of dose received by 5–95% of the GTV volume (GTV $D_{5\%}$–$D_{95\%}$).

## Results

### Stereotactic body radiation therapy planning

Fig 2 illustrates the axial, coronal, and sagittal views, along with dose-volume histograms, for the following SBRT plans: (a) ITV-SBRT and (b) gated-SBRT. As ITV-SBRT was planned based on all respiratory phases and gated-SBRT was based on phases the 30–70% respiratory phases, the GTV and PTV were smaller in the gated-SBRT plan (20.6 cc vs. 15.9 cc for GTV and 45.5 cc vs. 37.5 cc for PTV). However, as both plans followed the same optimization strategy, their dose-volume parameters remained comparable. Table 1 provides a summary of these parameters.

### Delivered dose analysis

When SBRT was delivered under the same respiratory motion conditions as the acquired 4D-CT set (amplitude: 10 mm, BPM: 20), both ITV-SBRT and gated-SBRT achieved adequate dose delivery. For ITV-SBRT, the average delivered doses at the superior edge, geometric center, and inferior edge were 616.1 cGy, 619.7 cGy, and 619.6 cGy, representing

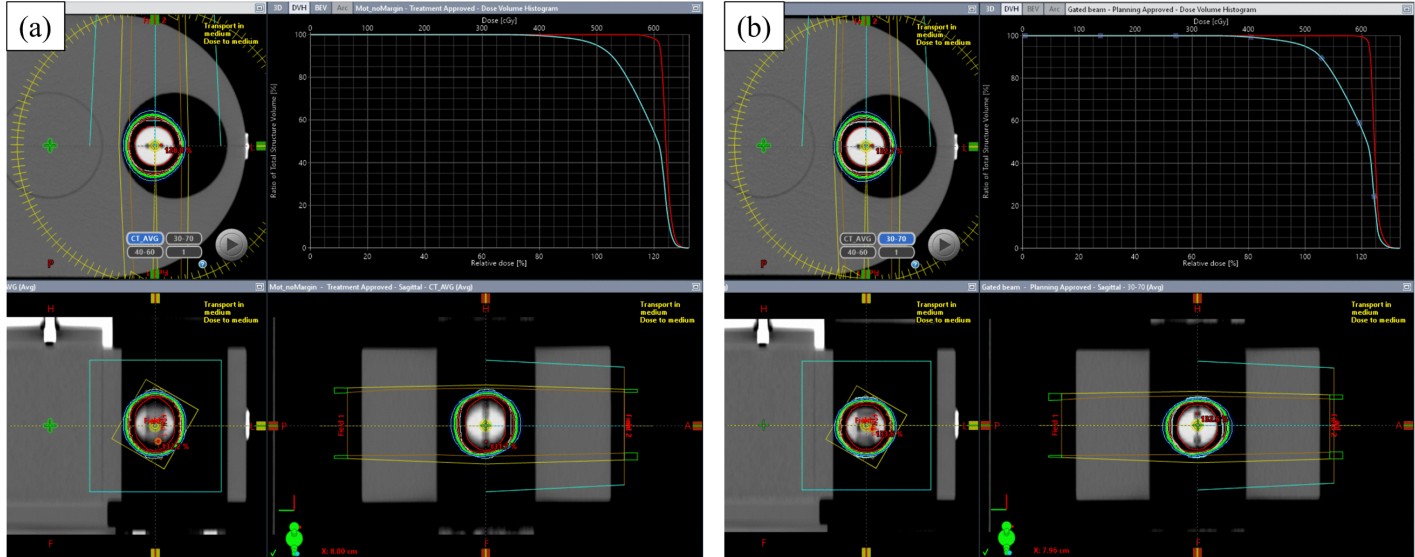

**Fig 2. Illustration of (a) created ITV-SBRT plan and (b) created gated-SBRT plan used in this study.** Each plan is presented in axial, coronal, and sagittal views, along with dose-volume histograms. ITV, internal target volume; SBRT, stereotactic body radiotherapy.

**Table 1. Dose-volume parameters for SBRT plans.**

|  | Parameters | ITV-SBRT | Gated-SBRT |
|---|---|---|---|
| GTV | Volume (cc) | 20.6 | 15.9 |
|  | $D_{max}$ (cGy) | 661.1 | 668.2 |
|  | $D_{mean}$ (cGy) | 620.8 | 623.3 |
|  | $D_{min}$ (cGy) | 562.4 | 581.5 |
|  | $D_{95\%}$ (cGy) | 609.4 | 613.2 |
|  | $D_{5\%}$ (cGy) | 636.1 | 638.2 |
| PTV | Volume (cc) | 45.5 | 37.5 |
|  | $D_{max}$ (cGy) | 661.1 | 668.2 |
|  | $D_{mean}$ (cGy) | 587.2 | 588.9 |
|  | $D_{min}$ (cGy) | 329.1 | 339.5 |
|  | $D_{95\%}$ (cGy) | 500 | 500 |
|  | $D_{5\%}$ (cGy) | 631.3 | 631.6 |

SBRT, stereotactic body radiotherapy; ITV, internal target volume;
GTV, gross tumor volume; PTV, planning target volume

99.2%, 99.8%, and 99.8% of the mean GTV dose, respectively, with a standard deviation of 3.9 cGy. The average delivered doses for gated-SBRT at the superior edge, geometric center, and inferior edge were 622.3 cGy, 618.6 cGy, and 613.5 cGy, representing 99.8%, 99.2%, and 98.4% of the mean GTV dose, respectively, with a standard deviation of 10.0 cGy (Table 2).

ITV-SBRT was delivered at three different breathing rates, BPM 10, 20, and 30, using the same 10 mm amplitude as in the 4D-CT acquisition to assess the influence of respiratory rate on dose delivery. The doses delivered to the superior edge, geometric center, and inferior edge of the target remained within the acceptable range, regardless of BPM variation. The average delivered doses for BPM 10 were 626.4 cGy, 620.4 cGy, and 618.2 cGy at the superior edge, geometric

**Table 2. Glass dosimeter dose verification results for SBRT plans under identical respiratory motion with 4D-CT acquisition.**

| SBRT | Location | Dose (cGy) | | | Average (cGy) | SD |
|------|----------|------------|------|------|---------------|-----|
| | | RT 1 | RT 2 | RT 3 | | |
| ITV | Superior | 610.2 (98.3%) | 622.2 (100.2%) | 616.1 (99.2%) | 616.1 (99.2%) | |
| | Center | 619.0 (99.7%) | 618.4 (99.6%) | 621.6 (100.1%) | 619.7 (99.8%) | 3.9 |
| | Inferior | 627.5 (100.1%) | 627.5 (99.3%) | 609.8 (100.0%) | 619.6 (99.8%) | |
| Gated | Superior | 624.4 (100.2%) | 635.8 (102.0%) | 606.5 (97.3%) | 622.3 (99.8%) | |
| | Center | 610.9 (98.0%) | 624.8 (100.2%) | 620.1 (99.5%) | 618.6 (99.2%) | 10 |
| | Inferior | 608.5 (97.6%) | 623.2 (100.0%) | 608.7 (97.7%) | 613.5 (98.4%) | |

SBRT, stereotactic body radiotherapy; ITV, internal target volume; CT, computed tomography; SD, standard deviation

center, and inferior edge, representing 100.1%, 99.9%, and 99.6% of the mean GTV dose, respectively, with a standard deviation of 5.8 cGy. For BPM 30, the doses were 621.6 cGy, 616.9 cGy, and 621.9 cGy at the superior edge, geometric center, and inferior edge, representing 100.1%, 99.4%, and 100.2% of the mean GTV dose, respectively, with a standard deviation of 7.0 cGy. Table 3 and Fig 3a illustrates these results.

ITV-SBRT was delivered at seven different respiratory amplitudes (amplitudes: 0–30 mm in 5 mm increments) while maintaining a constant BPM of 20 to assess the influence of respiratory amplitude on dose delivery for ITV-SBRT. The dose delivered to the geometric center of the target remained within the acceptable range, regardless of amplitude variations. However, dose delivery to the target edges gradually decreased as amplitude increased. When the amplitude exceeded 20 mm (i.e., > 10 mm beyond the reference amplitude), a significant underdose occurred, falling below the acceptable threshold ($D_{95\%} = 609.4$ cGy). For a 25 mm amplitude, the average doses delivered to the superior and inferior edges of the target were 587.1 cGy and 599.7 cGy, respectively, while for a 30 mm amplitude, they were 572.2 cGy and 593.7 cGy, respectively. The standard deviations increased progressively from 5.2 cGy at inferior amplitudes to 20.8 cGy at higher amplitudes. Table 3 and Fig 3b shows these results.

Table 4 and Fig 4 presents the delivered dose analysis for gated-SBRT. Treatment was delivered at five respiratory amplitudes (amplitudes: 10–30 mm in 5 mm increments) with a breathing rate of 20 BPM to assess the influence of amplitude variations on gated-SBRT. The dose delivered to the geometric center of the target remained within the acceptable threshold for all amplitudes, consistent with the ITV-SBRT findings. However, the dose distribution at the target edges differed from that observed in ITV-SBRT. In ITV-SBRT, increasing respiratory amplitude led to underdosing at both target edges. In gated-SBRT, the dose delivered to inferior edge of the target maintained an acceptable dose irrespective of changes in amplitude. However, the dose delivered to superior edge showed far significant underdose than that observed in ITV-SBRT. At amplitude 20 mm, the dose delivered to the superior edge was below $D_{min}$ (575.1 Gy). At amplitude 30 mm, the dose delivered to the superior edge was 307.9 Gy, which is below 50% of the mean GTV dose calculated in TPS (623.3 Gy).

## Discussion

In this study, two respiratory factors, BPM and respiratory amplitude, were adjusted to evaluate their influence on the accuracy of SBRT delivery. The results showed that BPM had a minimal effect on treatment accuracy, while changes in amplitude significantly affected SBRT dose distribution. Specifically, increasing amplitude led to underdosing, particularly

**Table 3. Glass dosimeter dose verification results for ITV-SBRT with different BPMs and amplitudes.**

| Variable | | Average dose (cGy) | | | SD |
|---|---|---|---|---|---|
| | | Superior | Center | Inferior | |
| BPM | 10 | 626.4 (100.9%) | 620.4 (99.9%) | 618.2 (99.6%) | 5.8 |
| | 20 | 616.1 (99.2%) | 619.7 (99.8%) | 619.6 (99.8%) | 3.9 |
| | 30 | 621.6 (100.1%) | 616.9 (99.4%) | 621.9 (100.2%) | 7.0 |
| Amplitude | 0 | 631.9 (101.8%) | 633.8 (102.1%) | 630.1 (101.5%) | 5.2 |
| | 5 | 617.5 (99.5%) | 619.6 (99.8%) | 615.7 (99.2%) | 7.6 |
| | 10 | 616.1 (99.2%) | 619.7 (99.8%) | 619.6 (99.8%) | 3.9 |
| | 15 | 615.1 (99.1%) | 625.3 (100.7%) | 626.4 (100.9%) | 6.5 |
| | 20 | 607.4 (97.8%) | 620.5 (99.9%) | 612.8 (98.7%) | 8.1 |
| | 25 | 587.1 (94.6%) | 616.0 (99.2%) | 599.7 (96.6%) | 13.4 |
| | 30 | 572.2 (92.2%) | 619.1 (99.7%) | 593.7 (95.6%) | 20.8 |

SBRT, stereotactic body radiotherapy; ITV, internal target volume; BPM, breaths per minute; SD, standard deviation

at the target edges. An increase of respiratory amplitude by ≥ 10 mm beyond the respiratory motion observed in the 4D-CT acquisition resulted in a significant underdose, falling below the acceptable dose range. The influence of amplitude was more pronounced in gated-SBRT than ITV-SBRT.

A direct comparison was made between ITV-SBRT and gated-SBRT, which were planned using different respiratory phase ranges (full phase for ITV and 30–70% respiratory phases for gating). This methodological difference is not a limitation but rather an intentional design that reflects the fundamental difference between two clinical strategies in lung SBRT motion management. The ITV-based approach is a 'motion-encompassing' strategy that aims to cover the entire range of tumor motion. In contrast, respiratory gating is a 'motion-limiting' strategy designed to minimize the irradiated volume by delivering the dose only during the most stable and reproducible portion of the respiratory cycle. Therefore, this comparison provides a direct evaluation of which distinct clinical strategy is more dosimetrically susceptible to unexpected respiratory motion changes.The minimal influence of BPM seems to be related to the SBRT planning technique used in this study. The DCA plan is generated using forward planning, where the multi-leaf collimator (MLC) conforms to the target shape. In a DCA plan, the MLC leaves do not obstruct the target, thereby minimizing concerns regarding the interplay effect [25,26]. As long as the target remained within the MLC aperture, variations in BPM did not compromise dose accuracy.

Volumetric modulated arc therapy (VMAT) is a widely used SBRT technique that utilizes inverse planning to simultaneously modulating the MLC, dose rate, and gantry speed. This approach enables precise dose delivery to the target while minimizing exposure to surrounding organs at risk [27–29]. In contrast to the DCA plan, BPM exhibited a more significant effect on the VMAT plan, with studies reporting dosimetric deviations of up to 20% [30–33]. The influence of BPM on VMAT-SBRT may differ from that on DCA-SBRT utilized in current study. However, this study did not assess the respiratory motion effects on VMAT-SBRT due to the physical volume occupied by the glass dosimeter used in this experiment

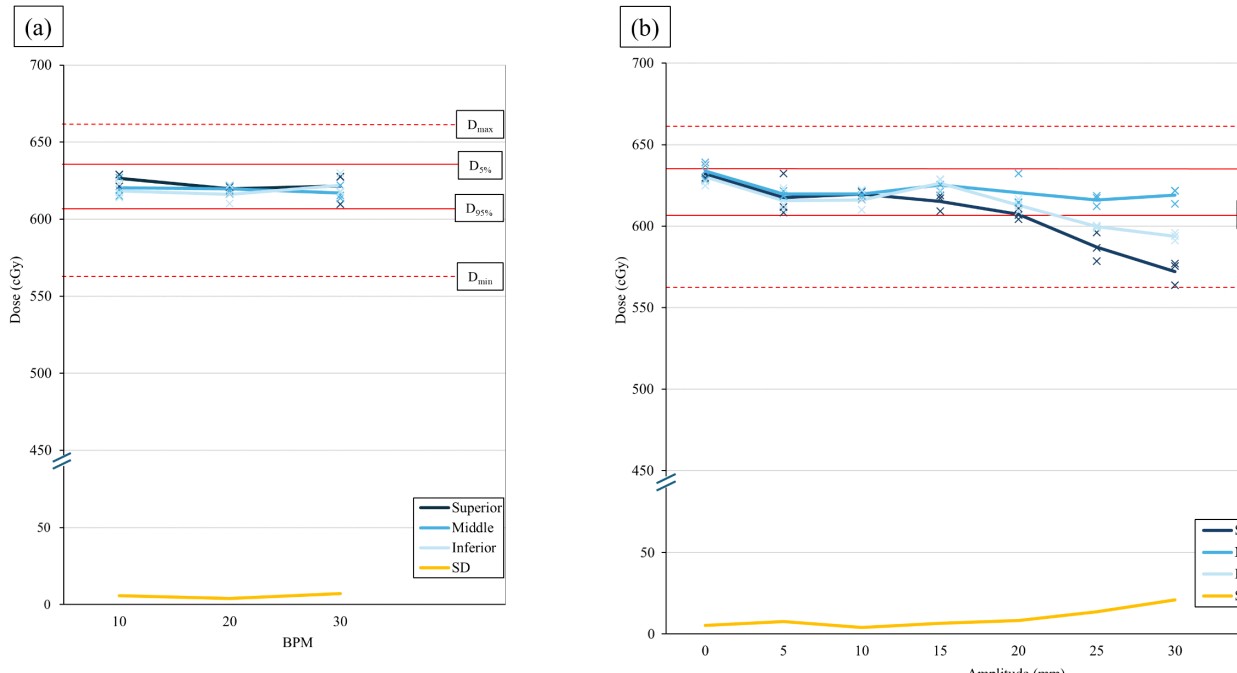

**Fig 3. Illustration of the influence of (a) BPM on ITV-SBRT, and (b) amplitude on ITV-SBRT to dose delivery.** Dose delivered to the superior edge (dark blue), center (blue), and inferior edge (light blue) of the target, along with their standard deviations (yellow). The average dose is represented by a line, while individual fraction doses are marked "x." Tolerance doses for $D_{5\%}$ and $D_{95\%}$ are represented by solid red lines, while $D_{max}$ and $D_{min}$ values are indicated by red dotted lines. BPM, breaths per minute; ITV, internal target volume; SBRT, stereotactic body radiotherapy.

**Table 4. Glass dosimeter dose verification results for gated-SBRT with different amplitudes.**

| Variable | | Average dose (cGy) | | | SD |
|---|---|---|---|---|---|
| | | **Superior** | **Center** | **Inferior** | |
| Amplitude | 10 | 622.3 (99.8%) | 618.6 (99.2%) | 613.5 (98.4%) | 10.0 |
| | 15 | 604.7 (97.0%) | 624.2 (100.1%) | 637.6 (102.3%) | 14.9 |
| | 20 | 575.1 (92.3%) | 623.9 (100.1%) | 632.6 (101.5%) | 27.2 |
| | 25 | 451.1 (72.4%) | 607.8 (97.5%) | 634.1 (101.7%) | 85.8 |
| | 30 | 307.9 (49.4%) | 610.9 (98.0%) | 624.2 (100.1%) | 155.0 |

SBRT, stereotactic body radiotherapy; SD, standard deviation

(Fig 1a). While glass dosimeter provides direct dose measurements at specific points, it is unreliable for VMAT when the target becomes occluded by the MLC during treatment. Fig 5 illustrates a simplified illustration of the limitations associated with using a glass dosimeter in a VMAT plan. During the RT session, the target received a total dose of 6 Gy; however, the glass dosimeter was fully exposed to 3 Gy and partially exposed to 6 Gy. Considering that partial exposure of the glass

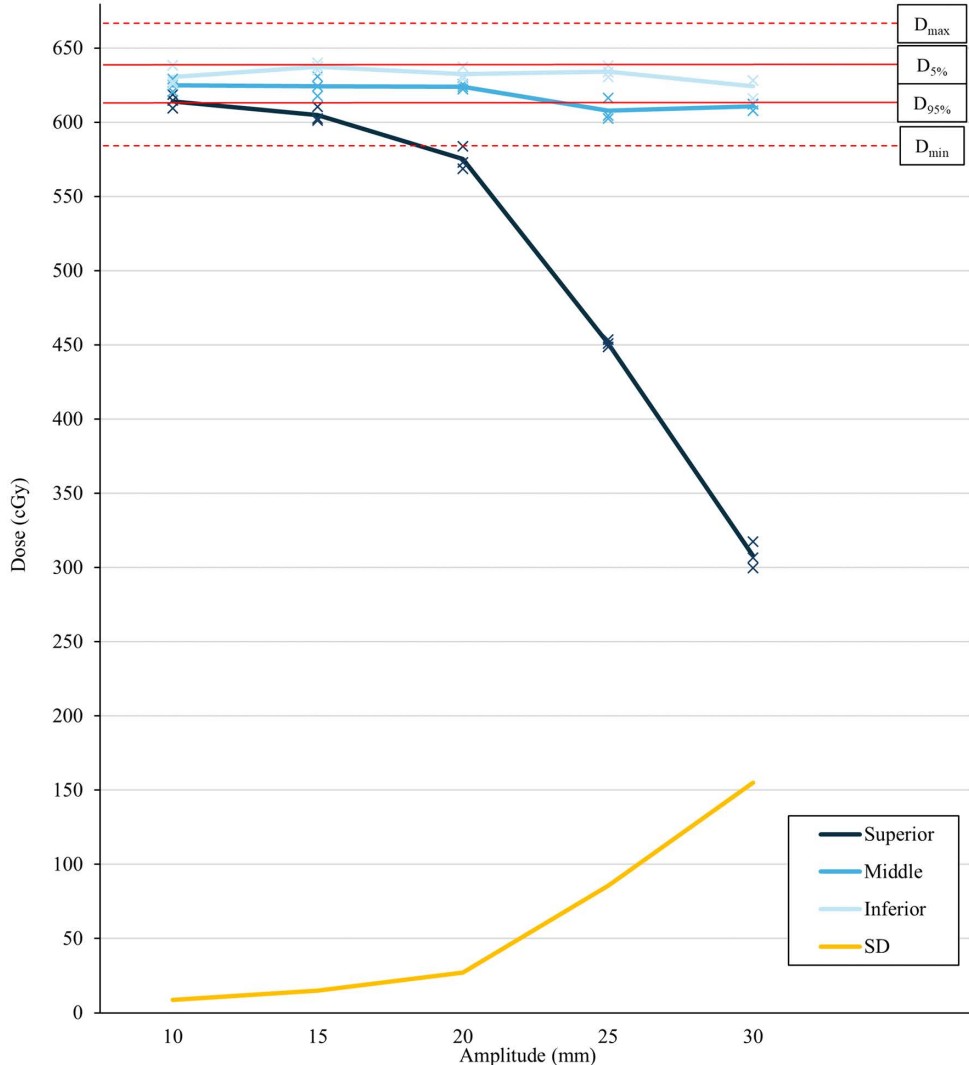

**Fig 4. Illustration of the influence of amplitude on gated-SBRT.** Dose delivered to the superior edge (dark blue), center (blue), and inferior edge (light blue) of the target, along with their standard deviations (yellow). The average dose is represented by a line, while individual fraction doses are marked "x." Tolerance doses for $D_{5\%}$ and $D_{95\%}$ are represented by solid red lines, while $D_{max}$ and $D_{min}$ values are indicated by red dotted lines. BPM, breaths per minute; ITV, internal target volume; SBRT, stereotactic body radiotherapy.

dosimeter cannot be accurately quantified, the dosimeter reading value is unreliable for accurately determining the dose delivered to the target.

When the respiratory amplitude during the RT session exceeds the value recorded in the planning CT scan, the target may shift beyond the MLC aperture, leading to underdosing. At the same respiratory amplitude, underdosing was more pronounced in gated-SBRT than in ITV-SBRT. Furthermore, while ITV-SBRT exhibited underdosing at both target edges, gated-SBRT resulted in more severe underdosing predominantly on one edge. Fig 6 illustrates this difference schematically. Fig 6a shows target movement throughout the respiratory cycle for a 10 mm amplitude. ITV-SBRT incorporates all respiratory phases during treatment (Fig 6b), while gated-SBRT uses only specific respiratory phases (Fig 6c). Accurate treatment delivery is achieved when the respiratory amplitude during SBRT matches that recorded in the planning CT. However, changes in respiratory amplitude during treatment can lead to significant dosimetric deviations. For example,

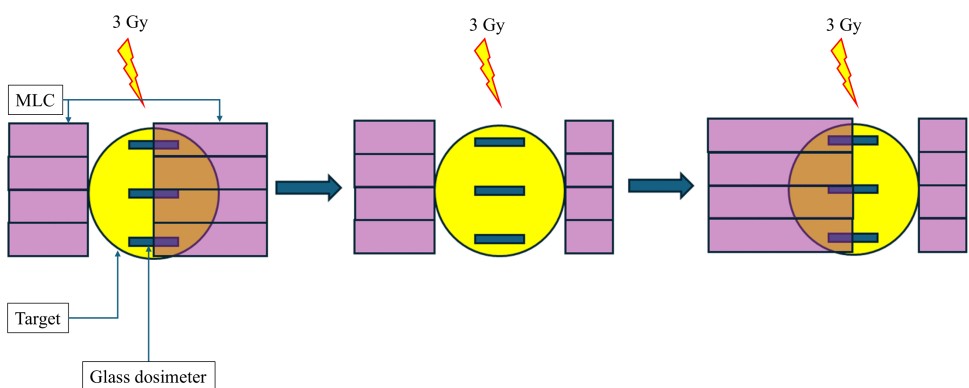

**Fig 5. Simplified illustration of the challenges associated with using a glass dosimeter in a VMAT plan, demonstrated with the step-and-shoot method.** While the target receives 6 Gy during the treatment session, the glass dosimeter is fully exposed to 3 Gy and partially exposed to 6 Gy. The actual VMAT plan is significantly more complex than this illustration, and the reading value of the glass dosimeter does not accurately represent the dose delivered to the target. VMAT, volumetric modulated arc therapy; MLC, multi-leaf collimator.

if SBRT is planned for a 10 mm amplitude but the target amplitude increases to 30 mm during treatment (Fig 6d), substantial changes in dose distribution may occur. In ITV-SBRT, an increase in amplitude causes both target edges to move beyond the MLC aperture for 30% of the beam-on time, respectively (Fig 6e). In contrast, in gated-SBRT, an increase in amplitude primarily displaces the superior edge of the target, causing it to remain outside the MLC aperture for 60% of the beam-on time. Furthermore, the inferior edge moves further inward within the MLC aperture, leading to unintended irradiation of the surrounding tissue near the inferior edge of the target (Fig 6f). This amplitude-induced discrepancy in dose coverage indicates that gated-SBRT is more sensitive to respiratory motion changes than ITV-SBRT.

Compared to ITV-SBRT, gated-SBRT offers a dosimetric advantage by reducing both the PTV and the volume of irradiated organs at risk (OARs) volume [34–38]. However, this dosimetric advantage is typically modest and does not always translate to clinically significant improvements [39,40], particularly considering the increased treatment time required [11,41]. Consequently, gated-SBRT is not routinely recommended and is instead reserved for carefully selected patients [36,40,42]. While no definitive clinical indications exist, gated-SBRT is generally considered for tumors exhibiting significant respiratory motion, tumors located near critical OARs, or patients with pre-existing pulmonary conditions, such as interstitial lung disease or chronic obstructive pulmonary disease [2,43]. Although these patients require strategies to minimize irradiated volume, they often exhibit irregular breathing patterns due to underlying disease or tumor-related factors [44]. As demonstrated in this study, irregular breathing results in inconsistent dose delivery, leading to target underdosing and overdosing of surrounding normal tissue, which may contribute to treatment failure or unexpected toxic effects. To ensure consistent and reproducible respiration, respiratory coaching should be conducted before CT acquisition and treatment delivery [16,45]. If reliable respiratory motion management cannot be achieved, alternative techniques, such as abdominal compression to minimize tumor motion, should be considered as an alternative to gated-SBRT [46,47].

Numerous studies have examined the relationship between respiratory motion and SBRT [34–36,38–40,42]. While the general influence of respiratory motion is known, this study provides specific, quantitative experimental data using point dosimetry on the impact of inter-fractional amplitude changes on dose delivery accuracy for lung SBRT. Identifying a potential dosimetric tolerance threshold, < 10 mm amplitude change from planning CT, offers practical information for motion management strategies and valuable insights into lung SBRT.

A key limitation of this study is the use of a DCA plan for lung SBRT. In clinical practice, VMAT-SBRT is more commonly used than DCA-SBRT [25,48]. Fig 5 demonstrates that the glass dosimeter used in this study does not accurately

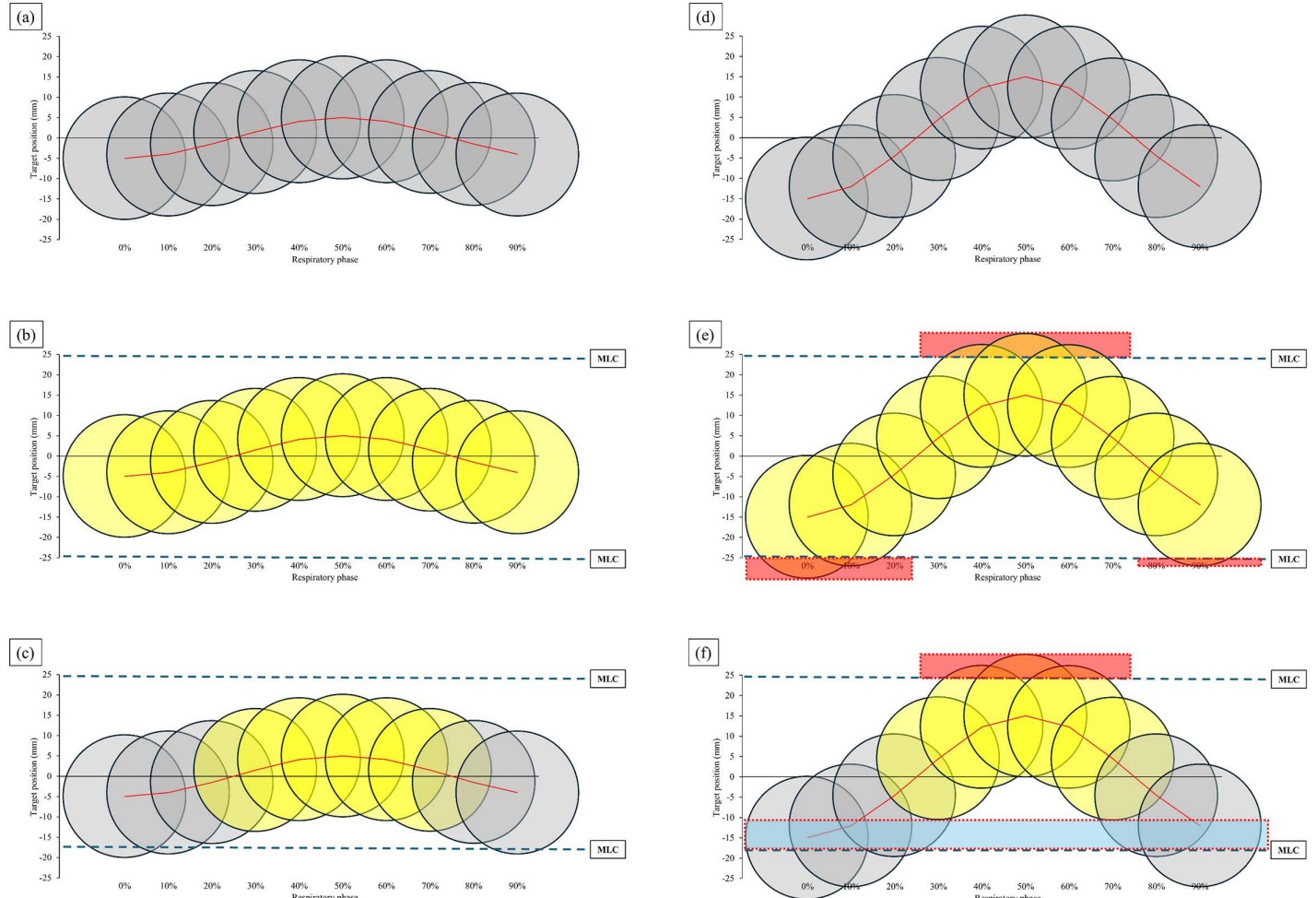

**Fig 6. Illustration of ITV-SBRT and gated-SBRT. (a)** Target movement throughout the respiratory cycle for a 10 mm amplitude. MLC location and beam-on phases for **(b)** ITV-SBRT and **(c)** gated-SBRT, with the target at beam-on phases highlighted in yellow. **(d)** Target movement for a 30 mm amplitude. **(e)** An increase in amplitude in ITV-SBRT causes the target to move outside the MLC aperture at both edges for 30% of the beam-on time. **(f)** An increase in amplitude for gated-SBRT results in the target moving outside the MLC aperture at the superior edge for 60% of the beam-on time and leads to unwanted normal tissue irradiation near the inferior edge. The target outside the MLC aperture is highlighted in red, while unwanted normal tissue irradiation is highlighted in blue. ITV, internal target volume; SBRT, stereotactic body radiotherapy; MLC, multi-leaf collimator.

measure the delivered dose in VMAT-SBRT. Considering that a VMAT plan involves more complex MLC motion than a DCA plan, we could speculate that the dose distribution in VMAT-SBRT would be more sensitive to respiratory motion changes. However, this hypothesis remains unverified in this study, necessitating further research to determine the acceptable respiratory motion range for VMAT-SBRT. Also, this study utilized sinusoidal waveforms and a QUASAR™ Respiratory Motion Phantom, which is restricted to motion in SI direction. Actual physiological respiratory motion is much more complex, involving multi-directional movements, rotation, and deformation [49]. While these simplifications limit the direct application of the quantitative findings from current study to clinical situations, its fundamental insights remain valuable. The observed greater sensitivity of gated-SBRT to respiratory motion variations over ITV-SBRT, will also be expected in actual clinical practice.

## Conclusion

Respiratory motion significantly affects dose delivery accuracy in lung SBRT, with amplitude playing a critical role. An increase in amplitude results in target underdosing, which is more pronounced in gated-SBRT than in ITV-SBRT. An amplitude increase of ≥ 10 mm from the respiratory motion observed during 4D-CT acquisition resulted in significant target underdosing, falling below the acceptable dose range. Effective respiratory motion management is essential for maintaining consistent, reproducible breathing patterns and ensuring accurate dose delivery.

## Supporting information

**S1 Table. Glass dosimeter calibration.**
(DOCX)

## Author contributions

**Conceptualization:** Bong Kyung Bae, Jae-Chul Kim.

**Data curation:** Bong Kyung Bae.

**Formal analysis:** Bong Kyung Bae, Sung Joon Kim.

**Investigation:** Bong Kyung Bae, Jae-Chul Kim.

**Methodology:** Bong Kyung Bae, Sung Joon Kim, Jae-Chul Kim.

**Project administration:** Bong Kyung Bae, Jae-Chul Kim.

**Supervision:** Jae-Chul Kim.

**Validation:** Bong Kyung Bae, Sung Joon Kim.

**Visualization:** Bong Kyung Bae.

**Writing – original draft:** Bong Kyung Bae.

**Writing – review & editing:** Bong Kyung Bae, Sung Joon Kim, Jae-Chul Kim.

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
