## [Decision Letter · Decision Letter 0]

30 Aug 2025

Dear Dr. Kim,

Thank you for submitting your manuscript to PLOS ONE. After careful consideration, we feel that it has merit but does not fully meet PLOS ONE’s publication criteria as it currently stands. Therefore, we invite you to submit a revised version of the manuscript that addresses the points raised during the review process.

We look forward to receiving your revised manuscript.

Kind regards,

Christopher Njeh

Academic Editor

PLOS ONE

**Journal Requirements:**

1. When submitting your revision, we need you to address these additional requirements. Please ensure that your manuscript meets PLOS ONE's style requirements, including those for file naming. The PLOS ONE style templates can be found at https://journals.plos.org/plosone/s/file?id=wjVg/PLOSOne_formatting_sample_main_body.pdf and https://journals.plos.org/plosone/s/file?id=ba62/PLOSOne_formatting_sample_title_authors_affiliations.pdf 2. In the online submission form, you indicated that “The data that support the finding of this study are available upon reasonable requests to the corresponding author.”  All PLOS journals now require all data underlying the findings described in their manuscript to be freely available to other researchers, either a. In a public repository, b. Within the manuscript itself, or c. Uploaded as supplementary information.This policy applies to all data except where public deposition would breach compliance with the protocol approved by your research ethics board. If your data cannot be made publicly available for ethical or legal reasons (e.g., public availability would compromise patient privacy), please explain your reasons on resubmission and your exemption request will be escalated for approval. 3. When completing the data availability statement of the submission form, you indicated that you will make your data available on acceptance. We strongly recommend all authors decide on a data sharing plan before acceptance, as the process can be lengthy and hold up publication timelines. Please note that, though access restrictions are acceptable now, your entire data will need to be made freely accessible if your manuscript is accepted for publication. This policy applies to all data except where public deposition would breach compliance with the protocol approved by your research ethics board. If you are unable to adhere to our open data policy, please kindly revise your statement to explain your reasoning and we will seek the editor's input on an exemption. Please be assured that, once you have provided your new statement, the assessment of your exemption will not hold up the peer review process. 4. Please include captions for your Supporting Information files at the end of your manuscript, and update any in-text citations to match accordingly. Please see our Supporting Information guidelines for more information: http://journals.plos.org/plosone/s/supporting-information. 5. If the reviewer comments include a recommendation to cite specific previously published works, please review and evaluate these publications to determine whether they are relevant and should be cited. There is no requirement to cite these works unless the editor has indicated otherwise. 

Reviewers' comments:

**Comments to the Author**

1. Is the manuscript technically sound, and do the data support the conclusions?

Reviewer #1: Yes

Reviewer #2: Yes

2. Has the statistical analysis been performed appropriately and rigorously?

Reviewer #1: Yes

Reviewer #2: Yes

3. Have the authors made all data underlying the findings in their manuscript fully available?

Reviewer #1: Yes

Reviewer #2: Yes

4. Is the manuscript presented in an intelligible fashion and written in standard English?

Reviewer #1: Yes

Reviewer #2: Yes

**Reviewer #1:**  Strength:

The study addresses a clinically important question about motion management in lung SBRT, specifically examining the effects of respiratory rate and amplitude on dose delivery using a 4DCT phantom and direct dosimetric measurements.

The authors should address the following questions to further strengthen the manuscript:

1. While direct measurement with a glass dosimeter adds robustness, please provide more detail on the calibration, linearity, energy dependence, and measurement uncertainties. Additionally, please clarify the readout process for the glass dosimeters, including how potential signal fading, background correction, or inter-dosimeter variability were addressed. Thank you for describing the dosimeter placements at the center and edges of the PTV if possible, please elaborate on whether these locations correspond to regions of highest dose gradient or areas most at risk of underdosing in clinical SBRT.

2. Given that the ITV was generated from the full phase, while the gated PTV used only the 30–70% phase range, could the authors discuss how this difference in phase selection may affect the comparability of the two approaches?

3. Given the importance of dose accuracy in SBRT, particularly at target edges, could the authors discuss why 2.5 mm was selected for the calculation grid size, and whether any assessment was made of the potential impact compared to a finer resolution?

**Reviewer #2:**  The manuscript was well written and the data was organized well in its scope. Here are some comments and suggestions:

1. Since the comparison of SBRT plans was done using DCA technique, but not VMAT. It's better to reflect this in the title, as majority of the SBRT plans were made using VMAT now.

2. The figures are really blurry and several labels can not be seen clearly.

3. In line 30, as these are DCA plans, the plans were not really "optimized". Consider to use the phrase "were created".

4. In line 36 and the following context, the term "upper and lower edges" were inaccurate. Please use "anterior and posterior edges" if it's in AP direction and "superior and inferior edges" if it's in SI direction.

5. In line 134, it is mentioned the resolution size is 2.5mm, which is a little bit big for SBRT plan. Please include some explanation for this.

6. In Figure 3 and Figure 4, there is no unit on y axis.

7. In line 178, Are the percentage values in the parenthesis the percentage of the dose delivered to these dosimeters in the plan. Please add some explanation to these values.

8. It would be clearer to include a table like table 2 to show what's the percentage dose change from the original values for different BPM and amplitude at different locations.

**Do you want your identity to be public for this peer review?** For information about this choice, including consent withdrawal, please see our Privacy Policy

Reviewer #1: No

Reviewer #2: No

---

## [Author Response · Author response to Decision Letter 1]

3 Sep 2025

Thank you for reviewing our manuscript. We have revised the manuscript according to journal requirements and the comments of reviewers. Point by point response to the reviewers are submitted separately as 'response to reviewers' file. We hope that the re-submitted version of the manuscript meets the high standards of the journal.

---

## [Decision Letter · Decision Letter 1]

27 Oct 2025

Influence of inter-fractional respiratory motion changes on dose delivery accuracy in dynamic conformal arc lung stereotactic body radiotherapy: A phantom study

PONE-D-25-24548R1

Dear Dr. Kim,

We’re pleased to inform you that your manuscript has been judged scientifically suitable for publication and will be formally accepted for publication once it meets all outstanding technical requirements.

Kind regards,

Christopher Njeh

Academic Editor

PLOS ONE

Additional Editor Comments (optional):

Reviewers' comments:

Reviewer's Responses to Questions

**Comments to the Author**

Reviewer #1: All comments have been addressed

Reviewer #3: All comments have been addressed

2. Is the manuscript technically sound, and do the data support the conclusions?

Reviewer #1: Yes

Reviewer #3: Yes

3. Has the statistical analysis been performed appropriately and rigorously?

Reviewer #1: Yes

Reviewer #3: N/A

4. Have the authors made all data underlying the findings in their manuscript fully available?

Reviewer #1: Yes

Reviewer #3: Yes

5. Is the manuscript presented in an intelligible fashion and written in standard English?

Reviewer #1: Yes

Reviewer #3: Yes

Reviewer #1: (No Response)

Reviewer #3: I am satisfied with the modifications and corrections made by the authors according to the Reviewers' comments.

**Do you want your identity to be public for this peer review?** For information about this choice, including consent withdrawal, please see our Privacy Policy

Reviewer #1: No

Reviewer #3: No

---

## [Editor Report · Acceptance letter]

PONE-D-25-24548R1

PLOS ONE

Dear Dr. Kim,

I'm pleased to inform you that your manuscript has been deemed suitable for publication in PLOS ONE. Congratulations! Your manuscript is now being handed over to our production team.

Kind regards,

on behalf of

MD Alessandra Castelluccia

Academic Editor

PLOS ONE